# Quantitative Analysis of the Impact of Public Policies on the Sustainable Development Goals through Budget Allocation and Indicators

**Raffaele Sisto** [1,2,3,]*[ ], **Javier García López** [1,2,3], **Alberto Quintanilla** [2], **Álvaro de Juanes** [2,4], **Dalia Mendoza** [2,3], **Julio Lumbreras** [5] **and Carlos Mataix** [3]

[1] Departamento de Ingeniería de Organización, Administración de Empresas y Estadística, Escuela Técnica Superior de Ingenieros Industriales, Universidad Politécnica de Madrid, 28006 Madrid, Spain; javier.garcialope@alumnos.upm.es

[2] Smart & City Solutions SL, Calle Blasco de Garay 61, 28015 Madrid, Spain; alberto.quintanilla@smartandcity.com (A.Q.); alvaro.dejuanes@smartandcity.com (Á.d.J.); dalia.mendoza@smartandcity.com (D.M.)

[3] Innovation and Technology for Development Centre, Technical University of Madrid, Av. Complutense s/n, 28040 Madrid, Spain; carlos.mataix@upm.es

[4] School of Economics and Business Administration, University of Alicante, c. San Vicente del Raspeig s/n, Building 31, 03690 San Vicente del Raspeig (Alicante), Spain

[5] Harvard Kennedy School, 79 John F. Kennedy St, Cambridge, MA 02138, USA; julio.lumbreras@hks.harvard.edu

*   Correspondence: raffaele.sisto@alumnos.upm.es

**Abstract:** Measuring the advances performed in the 2030 Agenda and the contribution of public policies remains a key issue. Budgets are acknowledged as one of the most powerful tools made available to administrations to push forward this contribution, and so several initiatives have risen to align budget items and Sustainable Development Goal (SDG) performance at all levels. The aim of this paper is to go beyond simple alignment and statistically analyze the interlinkages between budget and SDG achievement data. We have used the Spanish local administrations budget, together with indicators used to measure the 2030 Agenda goals at the same level, and computed a correlation test in order to find where budget allocation has an impact. We have then looked further into the relevant impacts to split them into direct and indirect. The research found ca. 25% of the budget items with relevant statistical links to the SDGs, with the SDGs 11 and 15 being the least impacted and SDGs 1, 4, 7, 8 and 16 the most connected ones. This research aims to set the bases of an evidence-based decision-support tool for a more efficient and sustainable policy design.

**Keywords:** Sustainable Development Goals; SDG; 2030 Agenda; budget; public policy; evaluation; indicators; meta-analysis; impact assessment

## 1. Introduction

The 2030 Agenda aims to define inclusive measures that promote sustainability and resilience [1]. These measures derived in 17 Sustainable Development Goals (SDGs), with 169 targets that integrate cross-cutting approaches throughout 2030, combining the three dimensions of sustainable development (economic, social, and environmental), and applying to developing and developed countries alike [2]. Integrating all three dimensions of development and having been signed by all 193 UN member states in 2015, SDGs represent a universal commitment towards sustainability.

As they have enlarged the definition of sustainability by integrating the environmental dimension of development (which had been perceived as left aside in the previous Millennium Development Goals), SDGs are also linked with climate action frameworks such as the Paris Climate Agreement (signed at COP 21 in Paris, on 12 December 2015, with the central aim to strengthen the global response to the threat of climate change by keeping a global temperature rise this century well below 2 degrees Celsius above pre-industrial levels and to pursue efforts to limit the temperature increase even further to 1.5 degrees Celsius (UNFCCC).) [2]. The European Union (EU), its institutions, and member states have played a key role in the adoption of the Agenda, the SDGs, and the Paris Agreement [2].

With multidisciplinary and multidimensional scopes, SDGs also set an ambitious path that will require investments of all kinds and from all sectors [3]. Despite this search for multi-stakeholder collaboration, national governments remain primarily responsible for realizing this transformation, embedding the SDGs into their policy planning [4]. Public budgets are powerful instruments to address any agenda. Conclusions from the latest UNDP (United Nations Development Programme) meetings agreed on mainstreaming SDGs into budgets as an urge to reach the completion of SDGs towards 2030 [5].

In this regard, there is an emerging key area in need of attention—how to translate the SDGs into budgets [6], and how may agents and countries cost out gaps to achieve the SDGs by mobilizing resources and budgeting for SDG activities [7]. The UNDP Bangkok Regional Hub defines three complementary pillars of SDG-integrated budgeting: Improving Strategic Allocative Function of Budgets via SDGs, Informing Budgets on SDGs, and Improved Effectiveness and Efficiency of Budgets in Achieving SDGs [8].

The previous three pillars, with a particular interest in evaluation for improved effectiveness and efficiency, emphasize the need addressed by this paper for an extensive evaluation of those areas in which budget allocation would have the most significant impact. As an increasing number of countries are considering integrating the SDGs into their budgeting processes [9], the evaluation of budget allocation impact becomes equally increasingly relevant, and it is, thus, the aim of this paper. Since the 2017 session of the High-level Political Forum (HLPF), 23 countries have mentioned ongoing measures to link the SDGs to the national budget, or that they had considered such action [4]. Although they are all recent and still open to evaluation and improvement, relevant examples of the previous include France, Mexico, and Spain.

Although focused on environmental and green budgeting, the case of France has been noticed by the international arena. Its last Environmental Budget Report (*Rapport sur l'impact environnemental dy budget de l'État* available in: https://www.budget.gouv.fr/files/uploads/extract/2021/PLF_2021/brochure_IEE.pdf), published in October 2020, presented a new classification of budgetary expenses and tax rates according to their impact on the environment and identification of public resources of environmental character. It became the first country in the world to carry out an environmental qualification of this type in the whole state-wide budget [10].

In the case of Mexico, a strong effort towards budget alignment with SDGs has been made, covering not only the environmental aspects but the three dimensions of sustainability. The Mexican government acknowledged that without a budget alignment effort, it is hard to know if they are investing correctly and to monitor the actions and their results [11]. In one of its Budget Transparency reports focused on investment in SDGs, the Ministry of Finance and Public Credit aimed to identify how national planning is linked with the long-term global agenda and to know which of the existing programs contribute to the achievement of each of the SDGs. This way, the Mexican government would have the necessary inputs to carry out long-term strategic planning, as well as to monitor their progress and results. According to the impact identification, public policy decisions and budget allocation may be made based on a diagnosis of how much is currently being invested in each of the SDGs and what actions are being taken [11].

Thirdly, the Government of Spain has presented within their State Budgets for 2021 (full budget report available in: https://www.sepg.pap.hacienda.gob.es/Presup/PGE2021Proyecto/MaestroTomos/

PGE-ROM/TomosSerieAzul.htm) a spending alignment with the SDGs for the first time. Although they have declared the methodology to be under development and open to changes, it is one that has been designed using a double-sided quantitative and qualitative approach to measure the budgetary effort that each spending policy made for the achievement of the SDGs [12]. Their analysis has been extended to 282 spending programs corresponding to 25 expenditure policies. The report acknowledges that the comprehensive conceptualization of SDGs contrasted to the specificity of budget items and policies makes it difficult to detect all possible alignments. Hence, it recognizes the importance of identifying direct or immediate effects produced by the actions of each of the participating agents—whose quantification could cost—as well as the indirect effects that will occur in the future extend beyond their immediate purpose, producing additional added value on the completion of SDGs [12].

*Direct and Indirect Impact*

Although the Spanish budget alignment to SDGs limits its scope to the policy direct impacts on SDGs, it does put out the need for identifying those impacts that cannot be catalogued as direct. This dichotomy has been brought up and used by various other countries in their alignment methodology as well. Nevertheless, the definition of direct and indirect impact varies from one methodology to the other, based on subjective and quantitative or terminological evidence [13,14].

On the other hand, the notion of alignment is often confused with the concept of impact [15,16]; alignment of a budget policy to a particular SDG does not necessarily mean the first is having a direct or indirect impact on the second one. Moreover, when an impact is indeed identified, this may not be immediately perceived as positive. Budget alignment to SDGs proven to have an impact must go through one further step of the analysis to determine if that impact, direct or indirect, is effectively positive or, contrary to what might be expected, results negative [17].

A lack of shared terminology and methodology has been identified from the contrast exercise between diverse budget-SDGs alignment methodologies [18]. In this context, the Sustainable Development Solutions Network (SDSN) and its national chapters have produced several reports to analyze the national and local achievement status of the SDG from specific indicators aligned to specific targets. As these reports set the first ground of database and indicators, though only expressing correlation between SDGs and their targets, they have been used as the basis for the present paper statistical study of budget alignment to SDGs [19–22]. Hence, this paper aims to set the ground floor for designing a tool that identifies the impact of public policy budget aligned to SDGs, using data-based analysis in the search of eliminating subjective and quantitative analysis variability, and applying the methodology to the Spanish 2021 Budgetary Policies as a study case.

## 2. Materials and Methods

### 2.1. Dataset Definition

The research database needs to combine budget data and a quantitative measure of SDG achievement, both applicable within the same administrative boundaries.

The Spanish SGFAL (General Secretariat for Local and Regional Financing (*Secretaría General de Financiación Autonómica y Local*)) presents the statistics related to the general budget of all local Spanish entities, and their execution. This information is prepared from the data provided by those entities to the Ministry, within the framework of current legislation. Budget data are classified by taking into account both the economic nature of the income and expenses and the purposes or objectives that the latter is intended to achieve, considering the criteria established in the budget structure applicable in each year. The SGFAL Database consists of a set of tables with optimized information, in this sense, the data from the financial account tables and the functional account tables referring to the execution of municipalities budgets for different years have been collected for our study up to the last year available—2018 [23].

Despite sustainability achievement data sources are quite heterogeneous, SDSN reports provide an extensive global portfolio of public reports with a standard and widely-used methodology based on indicators connected to the 17 Sustainable Development Goals. The SDSN chapter in Spain, *Red Española de Desarrollo Sostenible* (REDS) publishes the report: "*Los Objetivos de Desarrollo Sostenible en 100 ciudades españolas*", assessing approximately 100 Spanish cities with more than 80,000 inhabitants and all the provincial capitals. In the report, a majority of the nation-level indicators have been adapted or redefined to the municipal context, taking into account the development plans and strategies of each municipality [24]. In initial filtering, this report identified the most relevant indicators for urban ecosystems available in open, public databases, from a sustainable development point of view and grouped them by keywords in each SDGs. Although there is a small share of indicators at the province level, pondered to approximate the local values, the variables used to match the local administration boundaries were required by our research. Those indicators not directly related to any of the goals, or redundant, were discarded. The data for the report were processed using the methodology of the Sustainable Development Solutions Network (SDSN) [25], subsequently calculating a synthetic index for each SDG, which is included in the report results. In total, two releases of this report have been published up to this date (2018, 2020), and both have been included in our study. There are some differences between both releases concerning the indicators chosen for each SDG and the cities included, which could lead to interference in our results [24,26].

The dataset was built with the indicator data files published by REDS, along with the functional accounting table of local Spanish municipalities budget execution. The entries in each dataset were (non-standard) values of the synthetic SDG indexes for a given municipality and 1 year, and the *per capita* executed budget for each functional category.

## 2.2. Correlations Between Goals and Budgets

The aim of this paper is to identify the linkages between the budget program's allocation and the sustainable development situation that happens at the local level. We chose to use the aggregated SDG indexes instead of the individual indicators, thus encapsulating their heterogeneity, and performed the analysis separately for each year to avoid the differences between the report releases cited above. Through the use of standardized synthetic indexes for each Sustainable Development Goal, each observation in the sample supported a previously ranked value. Hence, the scores collected in the matrix in the SDG result part represent the situation of each municipality in the field of sustainable development.

The result of our calculations will be another matrix in which goals are related to the different budget programs by a correlation coefficient.

There are two important aspects to emphasize: first is that given the nature of the databases, there were some lost observations (particularly in SDG 14) or budget programs accounted as zero; on the other hand, as previously mentioned, we ranked result variables, making it possible to observe the existing linkages between the budgets of cities with a worse situation in sustainable development and those more advanced in this field. Consequently, the correlations used in this analysis corresponded to Pearson's bivariate correlations ($\rho$) to observe the proportional change due to the association between the change in a variable and its associated [27]. At an early stage, it was considered to use Spearman's analysis, which can capture the nonlinear correlation between the variables and is less sensitive to outliers [28]. Unfortunately, due to the above-mentioned partial incompleteness of the database, it became impossible to use this last method.

Therefore, the correlations that were considered were those that have a linear distribution with a typical significance level equal to or lower than 0.05. In addition, we considered those correlation coefficients above 0.2 or below −0.2. This filter reflects the need to impose a low discard criterion in order to find the largest number of indirect impacts, that by this very definition will show less relation than the direct ones. Thus, Pearson's $\rho$ value greater than 0.2 was considered to indicate a synergy

(positive linkage) between any budget program and a Sustainable Development Goal. A value under −0.2 would, in the opposite, point out a trade-off (negative linkage).

### 2.3. Model Definition: Thresholds and Interactions

After defining the quantitative nature of the methodology, it is necessary to establish the theoretical framework to which it leads. We sought to build the bases of an impact model, in which effort (the measure of the resources employed by the incumbent agents, incarnated in this case by budget data) is contrasted with the outcome (the measure of magnitudes directly related to the quality of life of citizens or the provision of services, in the form of sustainability indicators) [29]. Further on, our framework aimed to distinguish direct and indirect impact based on quantitative correlations. While direct impacts come from immediate relation in nature, object or terminology of a given budgetary policy and an SDG, indirect impacts come from correlations that are considered relevant with other SDGs that would not have an immediate relation.

The first ones are those linkages that have a straightforward approach to sustainable development, where the domain of the functional budget program matches the SDG. For instance, a potential correlation found between executed budget in functional program 31 'Health' and the aggregated score of SDG 3 ('Good health and wellbeing') index would have a straightforward connection with an easily understandable interpretation: a larger health budget will help improve the achievement of SDG 3. The correlation factor was thus proportional to the effort-outcome function.

Instead, those linkages jumping across different domains, or providing additional information to the previously assumed, were those classified as indirect impacts. Following up with our example above, a potential positive interaction between the program 31 'Health' and the score in SDG 4 (Quality education) index provided the opportunity to analyze the budgetary impact and its possible consequences from different, wider perspectives.

That is why a key part of the study was to align all budget programs to one or more Sustainable Development Goals, as this alignment provided us with the advantage of comparing the different impacts that we can find in the correlation matrix and split them into direct and indirect. The classification generated was based on the research of other Spanish administrations following the same idea, plus additional reflections resulting from that research. Cases used as input include the administrations of La Rioja, Badajoz, and the State Budget alignment methodology.

Since 2019, La Rioja has included in its Budget project a calculation of the budget alignment with SDGs. It states their aim to become leaders in the implementation and dissemination of Sustainable Development objectives and to assume a task of pedagogy of these for the whole of the autonomous region [30]. Although it does not have a results-oriented budget, Rioja has been working on a new approach to the definition of public expenditure, which has allowed concretizing the fulfillment of objectives in actions. In the case of La Rioja, the analysis comes down to the 169 SDG targets level, detailing which budgetary programs and expenditures have an impact on which SDG target. However, even if this provides a much more specific analysis than the one that can be made from Spanish National Budget alignment (limited to an SDG and budgetary policy level), there is still no identification of whether the alignment of expenditure to a target has a direct or indirect impact. Thus, the analysis remained broad as one budgetary policy or program may have an (undefined) impact in a more likely long list of targets.

On the other hand, the Province of Badajoz established its own methodology, aimed at aligning the SDGs with the budget of each management center of the institution according to its organic, economic, and program classification. Although Badajoz's methodology does integrate the classification of direct and indirect impacts, the definition of those that are indirect remains quite broad. They define "direct impact" as "those items that contribute directly to the achievement of some SDG", and "indirect impact" as those items that do not [31]. For calculation, the direct impact came from the result of adding up the total economic amounts identified from those actions or programs that have a direct impact on SDGs by their nature or object. The monetary total was divided between the

total budget to get the percentage of funds from the total budget hast have a direct impact in any SDG. Afterwards, the indirect impact budget results from, first, the extraction of the direct impact budget from the total budget, and second, the multiplication of it by the percentage of direct contribution calculated above. For the total percentage of budget contributing to SDGs, both direct and indirect impact monetary amounts are added up, divided between the Council's total budget, and multiplied by a hundred (under a simple 3-rule scheme). In summary, all of the budget that is considered not to have a direct impact on an SDG given its nature or object is automatically considered to have an indirect impact, with no further specificity.

The 2021 Spanish State Budget, as explained above, also approaches the alignment to the SDGs. As main contributions, the associated methodology considers direct contribution from a budgetary item to one or more SDGs when this match the main goal of the item; regardless if there are other minor goals (indirect contribution); it also defines a budgetary effort index that quantifies the amount of resources within a specific budgetary item that are aligned with each SDG, as a ratio of its total budget allocation. As a result, in order to classify direct and indirect impacts, we applied an alignment based on the Spanish State Budget, La Rioja Budget Alignment project, and, where not applicable, on the coherence between SDG target definition and budget items. Table 1 shows the budget programs alignment to SDGs.

**Table 1.** Budget programs alignment.

| Policy ID | Policy Name | SDG Alignment |
| --- | --- | --- |
| 11 | Public debt | |
| 130 | General Administration of Security and Civil Protection | 5, 11, 16 |
| 132 | Security and Public Order | 5, 11 |
| 133 | Transit and Parking Management | 3, 11 |
| 134 | Urban Mobility | 3, 11 |
| 135 | Civil Protection | 5, 11, 16 |
| 136 | Fire prevention and extinguishing service | 5, 11 |
| 150 | General Administration of Housing and Urbanism | 11 * |
| 151 | Urbanism: planning, management, execution and urban discipline | 11 * |
| 152 | Housing | 11 * |
| 153 | Public Roads | 9, 11 * |
| 160 | Sewerage | 6, 9, 11 * |
| 161 | Household drinking water supply | 6, 9, 11 * |
| 162 | Waste collection, management and treatment | 11, 12 * |
| 163 | Street cleaning | 11 * |
| 164 | Cemeteries and funeral services | 11 * |
| 165 | Public lighting | 7, 9, 11 * |
| 170 | General Administration of the Environment | 11, 15 * |
| 171 | Parks and Gardens | 11, 15 * |
| 172 | Environmental protection and enhancement | 13, 14, 15 * |
| 211 | Pensions | 1, 5, 8, 10 * |
| 221 | Other financial benefits for employees | 1, 5, 8, 10 * |
| 231 | Primary social care | 1, 4, 5, 10 |
| 241 | Employment promotion | 4, 8 |
| 311 | Public health protection | 3 |
| 312 | Hospitals, care services and health centers | 3 |
| 320 | General Administration for Education | 4, 12 |
| 321 | Establishment of pre-school and primary schools | 4 |
| 322 | Creation of secondary education centers | 4 |
| 323 | Operation of pre-schools, primary and special education centers | 4, 12 |
| 324 | Operation of secondary education centers | 4, 12 |
| 325 | Monitoring compliance of compulsory schooling | 4 |
| 326 | Complementary education services | 4, 12 |
| 327 | Promotion of citizen coexistence | 4, 10 |
| 330 | General Administration of Culture | 8, 11 |
| 332 | Libraries and Archives | 8, 11 |
| 333 | Cultural facilities and museums | 8, 11 |

**Table 1.** *Cont.*

| Policy ID | Policy Name | SDG Alignment |
|---|---|---|
| 334 | Cultural Promotion | 8, 11 |
| 336 | Protection and management of historical and artistic heritage | 8, 11 |
| 337 | Leisure facilities | 8, 11 |
| 338 | Popular festivities and celebrations | 8, 11 |
| 340 | General Administration of Sport | 3 |
| 341 | Promotion of Sportive activities | 3 |
| 342 | Sports Facilities | 11 |
| 410 | General Administration of Agriculture, Livestock, and Fisheries | 2, 8, 12 |
| 412 | Improvement of agricultural structures and production systems | 2, 8, 12 |
| 414 | Rural Development | 2, 8, 15 |
| 415 | Protection and development of fishery resources | 8, 12 |
| 419 | Other actions in agriculture, livestock and fisheries | 2, 8 |
| 420 | General Administration of Industry and Energy | 8, 9 |
| 422 | Industry | 9 |
| 423 | Mining | 9 |
| 425 | Energy | 7, 9 |
| 430 | General Administration of Trade, Tourism and Small and Medium Enterprises | 8 |
| 431 | Trade | 8 |
| 432 | Tourist information and promotion | 8 |
| 433 | Business Development | 8 |
| 439 | Other sectorial activities | 8 |
| 440 | General Administration of Transport | 3, 8, 11 |
| 441 | Passenger transportation | 3, 8, 11 |
| 442 | Transport Infrastructures | 3, 8, 11 |
| 443 | Transportation of goods | 3, 8, 11 |
| 450 | General Administration of Infrastructures | 9 * |
| 452 | Hydraulic Resources | 9 * |
| 453 | Roads | 9 * |
| 454 | Local roads | 11, 9 * |
| 459 | Other infrastructures | 9 * |
| 462 | Research and studies related to public services | 4, 8, 9 |
| 463 | Scientific, technical and applied research | 4, 8, 9 |
| 491 | Information Society | 9 |
| 492 | Knowledge Management | 9 |
| 493 | Consumer and user protection | 9, 12 |
| 912 | Governing bodies | 16 |
| 920 | General Administration | 1,17 |
| 922 | Coordination and institutional organization of local entities | 1,17 |
| 923 | Basic Information and Statistics | 1,17 |
| 924 | Citizen Participation | 1,17 |
| 925 | Attention to citizens | 1,17 |
| 926 | Internal Communications | 1,17 |
| 929 | Contingencies, transitional situations and implementation contingencies | 1,17 |
| 931 | Economic and tax policy | 16 |
| 932 | Management of the tax system | 16 |
| 933 | Heritage management | 16 |
| 934 | Debt and cash management | 16 |
| 941 | Transfers to Autonomous Communities | 16, 17 * |
| 942 | Transfers to Local Territorial Entities | 16, 17 * |
| 943 | Transfers to other Local Entities | 16, 17 * |
| 944 | Transfer to the State Administration | 16, 17 * |

* Alignment based on coherence elaborated by expert consensus.

An additional important part of the research was to time-bind such impacts. The interaction between budgets and their effective performance was intuitively not immediate; in other words, the consequences of implementing different budgets were visible many years later. Hence, the databases

collected were treated in different ways. Firstly, to observe the effect and synergies directly influenced by public budgets we used the 2016 budgetary variables to correlate with the 2018 sustainable development situation, assuming a time lag of 2 years, where the effects of different budgetary policies would presumably be perceived. Likewise, 2018 budget data were used for the correlation with 2020 sustainable development report. The final goal of this analysis was to find out the interaction between budget variations and the differences in the Sustainable Development situation that exist; that is, how an increase in a budget item influences the change in its linked SDGs (positively or negatively).

## 3. Results

The raw product of our research are two correlation matrices relating budgetary items (budgetary programs) with SDGs performance, the first one relating 2016 budgets and 2018 SDG indexes, and the other one 2018 budgets and 2020 indexes. The results obtained using the data available for the cities studied were analyzed using a two stages model: a first stage where the results of each calculated matrix are cleaned and analyzed individually, in order to obtain a clear picture of the correlation between budget programs and the SDG situation for each year analyzed by using only the significative results, and a second stage where the results of both matrices, once refined, are aggregated and compared in order to reflect possible coincidences between years. We have filtered the data according to the thresholds described in the methodology, and finally encapsulated the results to fit into Figure 1.

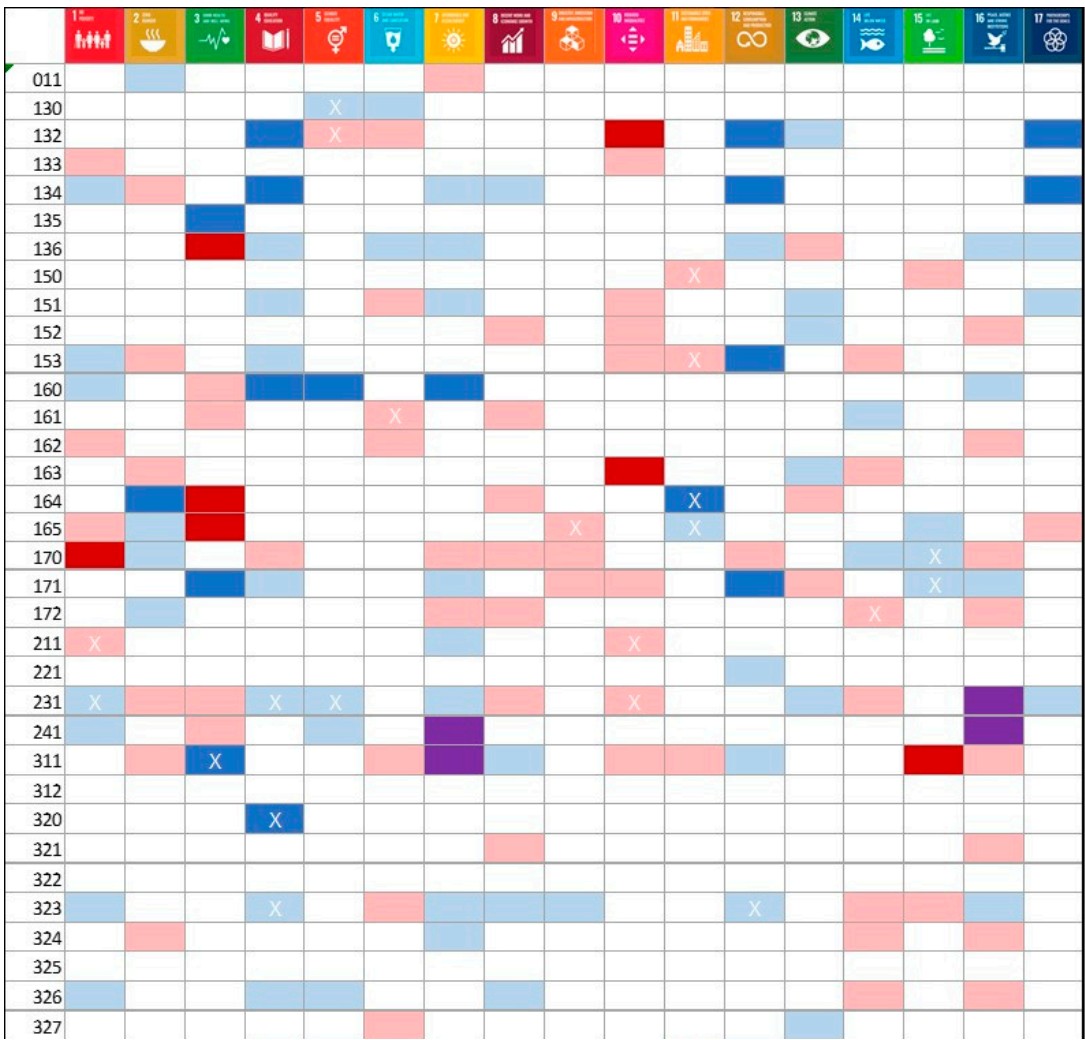

**Figure 1.** *Cont.*

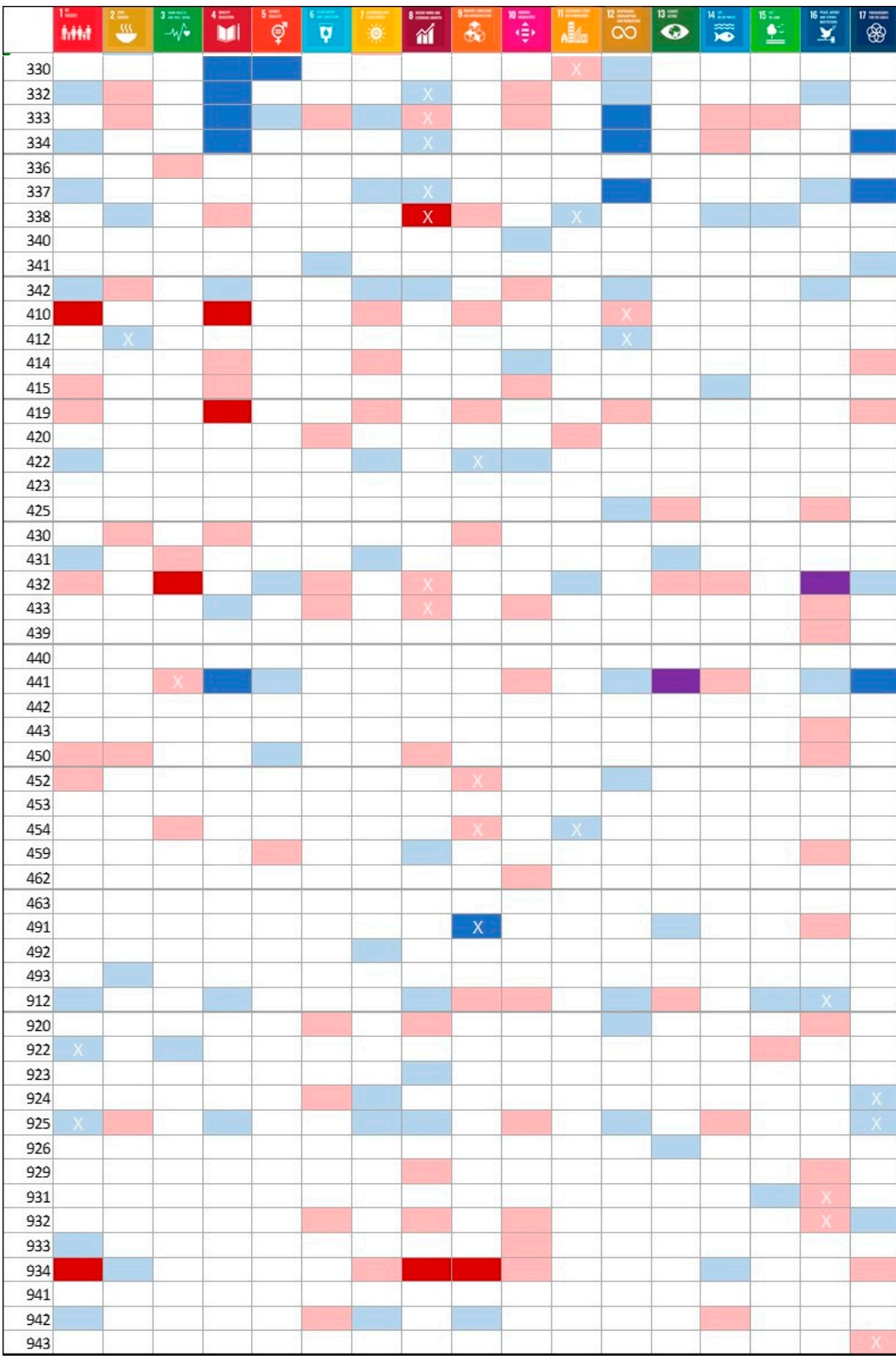

**Figure 1.** Summary of linkages found between budget items and Sustainable Development Goal (SDG) performance indexes. Red cells mean negative impact (trade-offs); blue cells positive impact (synergies); purple cells point out mixed impact (opposite impact direction between analysis). Darker colors render a consistent relation (present in both 2018 and 2020 analysis). Cells marked with x label direct impacts (there is a significant correlation present, and it matches the budget item SDG alignment).

The results indicate that almost 25% of all budgetary programs have an impact on the Sustainable Development Goals, with a level of significance of 0.05. Particularly, SDGs 1 (No poverty), SDGs 4 (Quality education), SDGs 7 (Affordable and clean energy), SDGs 8 (Decent work and economic growth), and SDG 16 (Peace, justice and strong institutions) show impact relations over 30% of the global budget, exhibiting a more robust relationship with the Sustainable Development Goals scores.

Thus, SDG 1 (No poverty) is impacted by 34.1% of the budget programs, reflecting 18 positive impacts in at least one of the years considered, 9 negative impacts of the same temporary nature and 3 prolonged negative impacts in both periods. To emphasize, there are some impacts calculated for this goal that are consistent with previously linkages mentioned, such as programs 231 (Primary social care) and 925 (Attention to citizens) which have a positive correlation in the model calculated and an evident relationship with the SDG 1.

In the same way, SDG 2 (Zero hunger) is impacted by 25% of the budget programs, reflecting 12 negative impacts in at least one of the years considered, nine positive impacts of the same temporary nature and three prolonged positive impacts in both periods. Following the same analysis, we can observe the program 412 (Improvement of agricultural structures and production systems) is positively linked in the correlation, in exactly the same way as they influence the SDG under discussion.

The SDG 3 (Good health and wellbeing for people) is impacted by 18.2% of the budget programs, reflecting eight negative impacts in at least one of the years considered, one positive impact of the same temporary nature, four prolonged negative impacts in both periods and three positive impacts reiterated in the 2-year periods. The program 311 (Public health protection) impacts directly.

The SDG 4 (Quality education) is impacted by 30.7% of the budget programs, reflecting 11 positive impacts in at least one of the years considered, five negative impacts of the same temporary nature, nine prolonged positive impacts in both periods and three negative impacts reiterated in the two-year periods. It is obvious that for this particular SDG, most of the programs included in the education or cultural related policies will have a positive impact; indeed the model predicts even seven positive correlations of this policy, five of them present in both periods; these are: 320 (General Administration for Education), 323 (Operation of pre-schools, primary and special education centres), 326 (Complementary education services), 330 (General Administration of Culture), 332 (Libraries and Archives), 333 (Cultural facilities and museums), and 334 (Cultural Promotion), thus identifying a goal with a remarkable performance with the generated model.

The next one, SDG 5 (Gender equality), is impacted by 13.6% of the budget programs, reflecting eight positive impacts in at least one of the years considered, two negative impacts of the same temporary nature and two prolonged positive impacts in both periods. The whole of the budget programs should and often do include a gender perspective, but likewise, the interpretation of the possible link between a budget and this particular SDG is complex, we would like to point out the following programs, which have a positive correlation: 130 (General Administration of Security and Civil Protection), 231 (Primary social care), and 326 (Complementary education services).

Following, SDG 6 (Clean water and sanitation) which is impacted by 20.5% of the budget programs, reflecting 15 negative impacts in at least one of the years considered and three positive impacts of the same temporary nature. This goal specifically, has a peculiar interpretation, which is that it contains a large number of negative correlations, among them are the programs 151 (Urbanism: planning, management, execution and urban discipline) and 161 (Household drinking water supply), these relationships can be interpreted as an unfavorable management of budgets towards this SDG, however, it is a study that must be taken in a more detailed and exhaustive way.

SDG 7 (Affordable and clean energy) is impacted by 30.7% of the budget programs, reflecting 17 positive impacts in at least one of the years considered, seven negative impacts of the same temporary nature, one prolonged positive impact in both periods and two of the rarely mixed impacts. This SDG is highly interconnected between the programs and the other SDGs, showing in the model has a positive correlation with the programs 422 (Industry) and 492 (Knowledge Management), but also, it is

important to underline the two mixed impacts observed in the programs 241 (Employment promotion) and 311 (Public health protection) which are still awaiting interpretation and discussion.

Another one, the SDG 8 (Decent work and economic growth) is impacted by 31.8% of the budget programs, reflecting 14 negative impacts in at least one of the years considered, 12 negative impacts of the same temporary nature and two prolonged negative impacts in both periods. This SDG is highly interconnected between the programs and the other SDGs, showing in the model a negative correlation with the programs 338 (Popular festivities and celebrations) and 934 (Debt and cash management), but also, it is important to underline positive correlations with the programs included in the policies 24 (education) and 34 (employment promotion).

The SDG 9 (Industry, innovation and infrastructure) is impacted by 17% of the budget programs, reflecting three positive impacts in at least one of the years considered, 10 negative impacts of the same temporary nature, one prolonged positive impact in both periods and one negative impact reiterated in the 2-year periods. This SDG is showing a positive correlation with the program 441 (Information society) and a negative correlation with the 934 (Debt and cash management), but also, it is interesting to underline negative correlations with the programs included in the policies 17 (Environment) and 45 (Administration of the infrastructures); these two relationships can be interpreted as unfavorable management of budgets towards this SDG or the existence of inherited problems in the municipal territory.

The SDG 10 (Reduced inequalities) is impacted by 28.4% of the budget programs, reflecting three positive impacts in at least one of the years considered, 20 negative impacts of the same temporary nature and two negative impacts reiterated in the 2-year periods. This SDG is showing several negative correlations especially with all the programs 13 (public services), 15 (urbanism), 23 (social promotion), and 93 (economy and tax policy). This may be due to the fact that the indicators of the SDG 10 have a strong economic bias and share an economic component with the budget programs indicators.

The SDG 11 (Sustainable cities and communities) is impacted by 11.4% of the budget programs, reflecting four positive impacts in at least one of the years considered, five negative impacts of the same temporary nature and one prolonged positive impact in both periods. This SDG has very low levels of correlation with the budget programs. It may seem strange that urban-level budget programs have low impact on the SDG of cities, but it should be noted that in the Spanish SDG Performance report all the indicators of the 17 SDG are scaled at the municipal level. Therefore, the indicators that are included in this SDG are measuring a very heterogeneous concepts within the urban context (with a specific stress only in pollution indicators), thus generating a very low relationship between them. Therefore, although there are slight positive links with urban public service spending programs, it is necessary to go much deeper into this SDG with more consistent data.

The SDG 12 (Responsible consumption and production) is impacted by 27.2% of the budget programs, reflecting 14 positive impacts in at least one of the years considered, three negative impacts of the same temporary nature and seven prolonged positive impacts in both periods, meaning this SDG is the one receiving the most synergies. On the one hand, there are impacts coming from investments that can be identified with improving infrastructure (132, 134, 153, 171, 337). On the other one, there is a string synergy too with cultural enhancements (333, 334). Negative impacts come from budget programs related with the primary sector (410, 419).

The SDG 13 (Climate Action) is impacted by 18.2% of the budget programs, reflecting nine positive impacts in at least one of the years considered, six negative impacts of the same temporary nature and one of the rare mixed impacts. Among the budgetary items impacting positively are some related with housing and urban planning (151, 152), urban services and environment (163) that are easily linked to emissions reduction at the local level. However, there are also less intuitive impacts from social care, urban coexistence or Information Society that indicate some correlation between life conditions and climate awareness. Negative impacts from tourism promotion and energy are also quite straightforward.

The SDG 14 (Life below water) is impacted by 20.4% of the budget programs, reflecting five positive impacts in at least one of the years considered and 13 negative impacts of the same temporary nature. The trade-off in the relationship between this SDG and one of its core budget programs, 172 (Environmental Protection and Enhancement) is very likely due an unexpected effect we will discuss further below, where negative correlation can appear where a strong need is identified by policymakers who allocate increasing financial resources to fix it. Besides that, this SDG receives mainly negative impacts.

The SDG 15 (Life on land) is impacted by 12.5% of the budget programs, one of the least connected, reflecting six positive impacts in at least one of the years considered, four negative impacts of the same temporary nature and one negative impact reiterated in the two year periods. There is a counterintuitive impact from budget program 311 (Public Health Protection), probably pointing out again the same paradox as in SDG 14. Regardless of this interaction, programs 170 (Environment) and 171 (Parks and Gardens) show expected positive results.

The SDG 16 (Peace, justice and strong institutions) is impacted by 35.2% of the budget programs, reflecting nine positive impacts in at least one of the years considered, 19 negative impacts of the same temporary nature and three mixed impacts. This SDG is very heterogeneous and hard to measure consistently, which could explain the contradictory result of programs 432 (Tourist information and promotion), 231 (primary social care), and 241 (employment promotion).

The SDG 17 (Partnerships for the goals) is impacted by 20.4% of the budget programs, reflecting eight positive impacts in at least one of the years considered, five negative impacts of the same temporary nature, and five prolonged positive impacts in both periods. It is worth noting the positive, consistent impact of the budget allocated for public safety (132, 134), culture (334), and leisure (337). Some budget programs promoting citizen commitment (924, 925) have also some impact, as expected.

## 4. Discussion

The correct implementation of the SDGs is the only way forward to address the global challenge of reaching human wellbeing, economic prosperity, and environmental protection. This research laid the foundations to create a process that can successfully estimate the impact on the SDGs of allocating funds into different budget items.

The analysis of interlinkages between budget programs and SDGs show interactions involving synergies and trade-offs that will greatly affect the future situation at the local level. This research shows that policymakers can no longer design strategies based on subjective assumptions, but they also need to include in the decision-making process the tools needed to achieve all sustainable development goals. We are assuming public, open indicators collected by third-parties are among such tools for they can be easily used for benchmarking and encapsulate part (if not most) of the policies' outcome.

Our research highlights the existence of typically more synergies than trade-offs within and among the budget and SDGs in most cities. The observed synergies show a broad compatibility between budget programs and SDG where progress in one goal or investment in one program can leverage the fulfillments of the other goals. However, as initial research, due to the lack of consolidated data, this paper has considered only the first-order connections. Further research should be carried out to extend this framework and enable the inclusion of subsequent orders of linkages.

The Spanish context has been used for the research because the SDG performance reports and the budget alignment initiatives are pioneer and the most comprehensive worldwide to this date at the municipality level. However, it looks mandatory to extend the methodology and add further contexts with similar available data, in order to reach more solid interlinkages and reduce coincidental correlation. In this sense, the SDSN indicators reports portfolio is the right material to start with.

Another aspect to consider is the granularity level of the SDG data. The current research has been performed attending to SDG level, without going deeper into SDG targets. However, the analysis could be repeated by splitting the SDG performance between targets following some guidelines in the SDSN reports. More precise interlinkages would be found, drawing an even more powerful tool

for policymakers. The same is applicable for budget items: our research ends at a second level of budgetary disaggregation, because it is the standard defined in the Spanish local budget common database, but it is possible to dive deeper into programs for a better result. Besides, we have chosen to research the relation between budget and SDG performance, but splitting into indicators again allows the analysis of the links among them, probably bringing up new landscapes of interactions.

After consolidating the interlinkages, it is critical for the purpose of research going further into detail, analyzing the mechanisms in the background. Some doubts can arise whether the linkage appears due to real impacts or they have been created by wise policy designers who detected issues in an area and are investing to revert them. This should be further researched by expanding the time series and, if possible, performing panel data analysis to determine if the interlinkage is consistent in time and there is an investment–impact effect, and not the opposite. Correlation does not imply causality; this means observed synergies and trade-offs between budget programs and SDG indicators could be independently related to another process driving both variables and resulting in correlations. However, in this case, as the study is realized in pairs in each of the 101 cities, the existence of a great number of synergies and trade-offs show that the relation is not appearing only for coincidence.

There are two key features showing the coherence of both the alignment and the analysis: there is usually a common direction in the correlations between budget items and SDGs within the same budget policy, thus confirming the causality of linkages; besides, there is a high level of correspondence between budget items alignment and their correlated indicators' alignment, pointing out that the main impacts land on the addressed domain by policies.

This tool can support decision making by helping prioritize such budget programs that have a strong positive influence on SDGs, taking into account both direct and indirect interactions. Additionally, the methodology highlights goals that will need specific policy and fund allocation, as they will not be indirectly helped by progress on other goals. The budget programs that affect the progress of some SDGs negatively can be identified, and potential trade-offs and negative spill-over effects mitigated and anticipated. This approach facilitates dialogue between municipal policy makers from different sectors or departments, fostering the indirect impacts that one program could have over other goals. Beyond the public administration, discussions can appear with other stakeholders around different pathways and lead to other action areas that enhance progress on the 2030 Agenda achievement. Our approach offers leaders a connected, systemic view of the Agenda, and a common data-based language. This evidence-based analytical approach should drive to a quicker, more participated, and robust roadmap towards sustainable development.

The results of our analysis provide the basis for forthcoming researches that will be focused on the automatization of synergies and trade-off models for the main patterns identified. This would allow us to evaluate if the identified synergies are scalable and replicable to other levels.

**Author Contributions:** Conceptualization, R.S., J.G.L., and A.Q.; methodology, R.S. and A.Q.; validation, J.L. and C.M.; formal analysis, R.S., A.Q., and Á.d.J.; investigation, R.S. and A.Q.; resources, R.S., J.G.L., and D.M.; data curation, A.d.J.; writing—original draft preparation, R.S. and A.Q.; writing—review and editing, R.S., J.G.L., A.Q., Á.d.J., D.M., and J.L.; visualization, D.M.; supervision, R.S., J.L., and C.M.; project administration, R.S. All authors have read and agreed to the published version of the manuscript.

**Funding:** This research received no external funding.

**Conflicts of Interest:** The authors declare no conflict of interest.

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
