# Peer review of "Quantitative Analysis of the Impact of Public Policies on the Sustainable Development Goals through Budget Allocation and Indicators"

_sustainability, doi:10.3390/su122410583_

Round 1

Reviewer 1 Report

This is a solid paper devoted to the pertinent issue of the relationship between Agenda 2030 and public policies. The paper focuses on the analysis of the interlinkages between budget and SDG achievement in the case of Spain 2021 Budgetary Policies of 101 cities.

 I suggest to clarify the objectives of the paper at the beginning of the introduction, while now I have to get to lines 107 and 147 and following.

Author Response

Thank you for the comments and your review.

Response to Reviewer 1 Comments

Point 1: I suggest to clarify the objectives of the paper at the beginning of the introduction, while now I have to get to lines 107 and 147 and following.   Response 1: We insert a description of the aim of the paper and the objective between lines 57 and 66.   English language and style: After your suggestion, we reviewed the paper with an English native speaker.

Reviewer 2 Report

Attached.

Author Response

Thank you for your review, it is greatly useful and mind-opening for us. System dynamic approach is very inspiring and we will consider it in the future.

Point 1: I believe the discussion section could be strengthened to emphasize the challenges in the decision-making processes by the policymakers. Too often these decisions are fraught with bias and bounded rationality. 

And finally, the discussion section could use more clarity. While in theory, you may be making headway by tracking budgets and indicators. However, are the indicators tied to true measurable and sustainable outcomes, perhaps maximizing effect? See lines 409-410, 415-416. 

Response 1: Between lines 417 and 419 we inserted a comment based on your suggestions addressing the link between policies and the outcome measure using indicators.

Point 2: Lines 417-461 could use clarification or a more concrete discussion of the concepts and your recommendations. Fortifying your recommendation with limitations may help. For instance, there may be a causality or correlation with linkages but is significant headway being made and measured for true and meaningful outcomes? How will these outcomes be measured? 

Response 2: Inspired by your comment we have added a new key feature of the coherence between alignment and correlation (lines 452-456). We have observed a close relationship between SDG alignment in budget items and related indicators.

Concerning the limitations, we initially considered as boundaries the municipal competency as a boundary for the study but we decided not to, as SDG are a multilevel topic that has to be measured holistically.

English language and style: we applied English improvements to the paper with the help of an English native speaker.